# Effect of Acellular Dermal Matrix in Postoperative Outcomes in Tissue Expander Breast Reconstruction After Immediate Mastectomy

**DOI:** 10.3390/cancers17193185

**Published:** 2025-09-30

**Authors:** Óscar Nova-Tayant, Eduardo Saorín-Gascón, Ramón A. Moreno-Villalba, María A. Mora-Ortiz, Clemente J. Fernández-Pascual, Pablo J. Vera-García, Antonio Piñero-Madrona

**Affiliations:** 1Department of Plastic and Reconstructive Surgery, Hospital Virgen de la Arrixaca, 30120 Murcia, Spain; oscar.nova@um.es (Ó.N.-T.); eduardo.saorin@um.es (E.S.-G.); asuncionmoraortiz@gmail.com (M.A.M.-O.); clemente.fernandez@um.es (C.J.F.-P.); pablojvg2407@gmail.com (P.J.V.-G.); 2Breast Unit Surgery, Hospital Virgen de la Arrixaca, Ctra. Madrid-Cartagena, s/n, El Palmar, 30120 Murcia, Spain; morenovill@hotmail.com

**Keywords:** breast reconstruction, tissue expander, dermal matrix, postoperative outcomes, complications

## Abstract

**Simple Summary:**

Post-mastectomy breast reconstruction is currently considered an integral component of breast cancer treatment. Among the most significant advances in recent decades is reconstruction utilizing breast expanders and acellular dermal matrices, employed to improve the quality of the breast flap following mastectomy. However, despite the increasing use of these matrices, concerns regarding their safety profile have been raised due to reported complications in several studies The present study aims to analyze the outcomes associated with these matrices in breast expander reconstruction at different anatomical planes.

**Abstract:**

**Background:** Breast reconstruction following mastectomy has become an essential procedure in breast cancer treatment due to its positive impact on patients’ quality of life. Among the various reconstruction techniques, the use of expanders followed by implants has gained popularity. In this context, acellular dermal matrices (ADM) have been introduced as an adjunct to improve implant coverage, lower pole support, and aesthetic outcomes. However, their use has also been associated with higher costs and a potential increase in postoperative complications, which remains a matter of debate. We aimed to determine the relationship between acellular dermal matrix and postoperative outcomes and complications. Methods: An observational retrospective study was conducted with patients who underwent immediately breast mastectomy followed by tissue expander reconstruction from January 2022 to June 2024. Patients were divided into two groups depending on reconstructive plane. **Results**: The final cohort contained 87 patients. Smoking, radiotherapy and dermal matrix were associated with a higher complication rates. After risk-adjustment, dermal matrix use led to a higher rates of surgical site infection (OR 7.62, *p* = 0.029) in the prepectoral plane, and higher rates of overall complications (OR 3.34, *p* = 0.05) and surgical wound dehiscence (OR 6.04, *p* = 0.048) in the retropectoral plane. **Conclusions**: These findings highlight the importance of individualized surgical planning, particularly concerning the use of acellular dermal matrix, which were associated with increased risks of surgical site infection, dehiscence, and global complications. Further research is required to establish standardized guidelines for the optimal selection surgical technique.

## 1. Introduction

Post-mastectomy tissue expander breast reconstruction is a well-established surgical technique, increasingly recommended in the surgical treatment of breast cancer in some cases of lack of skin or radiotherapy adjuvant treatment, owing to its positive impact on patients’ quality of life and satisfaction and low complication rates [1,2,3,4].

Acellular dermal matrices (ADM) have been used during the last decade in reconstructive breast surgery. They were first employed in breast reconstruction in 2001 to reduce rippling in the upper pole resulting from breast implant placement in patients with thin mastectomy flaps.

These are biosynthetic materials composed of dermal or collagen components of various origins (human, porcine, or bovine) that integrate into the breast flap, allowing a complete breast pocket, providing coverage to the implant and increasing flap thickness, thereby preventing implant malposition and providing better definition of the inframammary fold [5,6]. Among its advantages, it also reduces capsular contracture rates by decreasing local inflammation [7].

Although numerous benefits have been reported, outcomes and complications obtained with these ADM are usually controversial. In this way, seroma, infection, skin flap necrosis and dehiscence have been associated with them [8]. This may be attributed to the retrospective nature of most studies, small sample sizes and the wide variety of products available on the market. However, evidence suggests that using these matrices is associated with lower rates of capsular contracture, albeit with a higher incidence of seroma formation and infection [9].

However, the complication rate in studies varies widely, ranging from 3.2% to 48.7%, with very disparate and often inconclusive results in the literature. Unfortunately, there are very few studies comparing the different types of matrices in terms of differences in their manufacture, origin, thickness, and type of processing, which is probably one of the main reasons for the discrepancy in the results found in the literature [10].

Although their application has become more prevalent since their introduction, particularly in immediate breast reconstruction with implants, their use in immediate-delayed breast reconstruction with tissue expanders has also increased in recent years. Its advantages are the same as those of breast implants, in addition to the possibility of greater intraoperative expansion, reducing the time to complete expansion and, therefore, accelerating the time to replacement with a permanent implant [11].

Despite these advances, standardized guidelines for selecting the appropriate surgical technique remain lacking, leading to significant inter- and intraindividual variability in clinical practice. This study aims to analyze the outcomes and complications following two-staged breast reconstruction according to reconstructive plane and acellular matrix use.

### 1.1. Patients and Methods

An observational retrospective study included all patients undergoing mastectomy followed by two-stage breast reconstruction with a tissue expander, with a minimum postoperative follow-up of six months, between January 2022 and June 2024. Patients were excluded if they underwent delayed reconstruction, immediate reconstruction with direct implant placement, or autologous tissue reconstruction (DIEP or latissimus dorsi flap). Also excluded were cases where tissue expanders were used as a salvage procedure following failed reconstruction (mainly due to failed microsurgical procedures). Lastly, patients were excluded if at least 80% of the data for the study variables could not be obtained.

Clinical, epidemiological, histopathological, and surgical data were collected from the patient’s medical record.

Complications were categorized as seroma, skin or nipple-areolar complex (NAC) necrosis, infection, and dehiscence. Seromas were recorded by clinical or ultrasonography signs. Necrosis included all cases with obvious skin necrosis which needed some intervention (oral or intravenous antibiotics, surgical debridement). Surgical site infection (SSI) was defined as a clinical situation presenting local signs such as erythema, increased exudate or local heat, with or without systemic symptoms (fever) and/or laboratory abnormalities (leukocytosis or elevated acute phase reactants) or positive bacterial cultures. Dehiscence included all cases with a disruption of the surgical wound that involved exposure of the implant requiring a return to the operating room to address the complication, often involving implant removal (with or without replacement during the same procedure).

The surgical approach was determined in advance by the Breast Hospital Committee.

Indication for two-stage reconstruction was a radical mastectomy with lack of skin coverage or axillary involvement requiring adjuvant radiotherapy.

The research project was presented and approved by the Ethics Committee for Clinical Research of the hospital.

The surgical procedure was performed by two surgical teams. Breast surgeons performed the mastectomy, sentinel lymph node biopsy (SLNB), and axillary lymphadenectomy (if indicated), while plastic surgeons performed the breast reconstruction. Lymphadenectomy was indicated in cases of axillary involvement confirmed by prior biopsy or the presence of macrometastases in the SLNB. The surgical plane for reconstruction was selected based on the characteristics of the mastectomy flap. The retropectoral plane was chosen when visible dermis remained after mastectomy or when there were signs of compromised mastectomy skin flap (erythematous discoloration, poor capillary refill, or absence of bleeding from the dermal edges). In all other cases, a prepectoral plane was preferred. No specific criteria were followed for the use of ADM and their use was determined according to the preferences of the surgical team. Reconstructions were performed by four plastic surgeons, all operating from the same patient pool. Two surgeons routinely employed ADM in nearly all reconstructions, whereas the other two generally avoided it and preferred muscle-only coverage or subcutaneous expander placement. Importantly, patients were not triaged differently among surgeons: case allocation was random within the unit, and the indication to use or not use ADM was made prior to intraoperative assessment of flap quality, rather than being reserved for more complex or compromised cases.

FORTIVA ^®^ (RTI Surgical, Neunkirchen, Germany) is a ready to use porcine dermis derived sheet, that does not require rehydration. It was preferred for prepectoral placement due to its larger size and thickness (1 mm), providing complete coverage of the implant and improved stabilization of its anatomical limits. TUTOMESH^®^ (RTI Surgical, Neunkirchen, Germany), an acellular matrix derived from bovine pericardium, mainly composed of collagen, was used in retropectoral plane because of its thinner profile (0.6 mm) and smaller dimensions, which allowed for adequate coverage of the inferolateral portion of the expander, with the rest of the expander covered by the pectoralis major muscle.

All cases employed anatomically shaped microtextured tissue expanders. The expander volume was determined primarily based on the breast base width. When tissue expander was used for axillary involvement and adjuvant radiotherapy, with an adequate mastectomy flap, it was filled intraoperatively to its full volume. Those patients with insufficient skin coverage. The expander was filled to a volume that allowed for tension-free wound closure.

Intravenous antibiotics were used routinely (cefazolin 2 g at the beginning of the procedure). All expanders, acellular matrices (when used), and mastectomy pockets were irrigated in a solution of normal saline, 80 mg of gentamicin, and 500 mg of vancomycin to reduce the risk of infection.

In FORTIVA^®^ placement, the ADM was shaped according to the tissue expander and sutured to the breast pocket using 3/0 absorbable monofilament at the superior and medial edges.

In case of TUTOMESH^®^, it was sutured with 3/0 absorbable monofilament to the inferolateral border of the pectoralis major and the anterior axillary line.

A 15-Fr Blake drain was placed in all cases. In procedures where an ADM was used, a second 10-Fr Blake drain was placed in front of the matrix. Drain removal was performed when the output was less than 30 mL/24 h for two consecutive days.

Two weeks after surgery, once the surgical wounds were completely healed, breast expansion began. Expansion was performed by injecting 30–50 cc per session, according to patient tolerance (considering skin tension, cutaneous pallor, or patient discomfort). Expansion sessions were typically scheduled every two weeks. Subsequently, RT was administered in the indicated cases. After a period of approximately 9–12 months, the expander was replaced with the final prosthesis.

### 1.2. Statistical Analysis

A convenience sampling method was employed, selecting participants based on their accessibility and availability.

Patients were retrospectively classified into two cohorts based on the reconstruction plane used (prepectoral vs. retropectoral). The primary outcome variable was the presence of complications.

A descriptive analysis was conducted to verify the normal distribution of quantitative variables using the Kolmogorov–Smirnov test. Normally distributed variables were described using the mean and standard deviation, while non-normally distributed variables were described using the median and range. Qualitative variables were described using absolute numbers and percentages.

A bivariate analysis was conducted using Student’s t-test for the association of quantitative and qualitative variables. The Mann–Whitney U test was used for non-parametric variables. The Chi-square test was used for the association of qualitative variables, with Fisher’s exact test used when the expected frequency count was below 20%. To avoid the effect of any potential confounding factors, a multivariate analysis was performed. (logistic regression).

The level of statistical significance was set at *p* < 0.05, and all statistical analyses were conducted using SPSS software (version 25).

The article has been written in accordance with the STROBE guidelines.

## 2. Results

A total of 87 breast reconstructions were analyzed. The median age at the time of surgery was 50 years. Up to one-third of the patients (31 cases) had a history of smoking. The predominant histological lesion was invasive ductal carcinoma. Post-mastectomy radiotherapy was administered to 39% of the patients, and up to 80% received hormone therapy after the procedure. Approximately one-fifth of patients (17.2%) received neoadjuvant chemotherapy (NACT), while almost half (39.1%) received adjuvant chemotherapy (ACT). Regarding surgical technique, ADMs were not utilized in slightly more than half of the cases.

The distribution was homogeneous with respect to the type of reconstructive plane, with a slight predominance of the retropectoral plane (37.9% prepectoral, 62.1% retropectoral).

Regarding the characteristics of the series according to the reconstructive plane, they were quite similar, differences were found only in the proportion of patients who received NACT (*p* = 0.012). The rest of the general characteristics of the series are summarized in Table 1.

In prepectoral plane, smoking was associated with a higher rate of overall complications (*p* = 0.05) and SSI (*p* = 0.05). Additionally, the use of hormonal therapy was associated with a higher rate of overall complications (*p* = 0.03).

In the retropectoral plane, an association was found between smoking and SSI (*p* = 0.04), as well as administration of radiotherapy and higher rates of SSI (*p* = 0.05). Furthermore, lymph node involvement was associated with a higher rate of overall complications (*p* = 0.01) and surgical wound dehiscence (*p* = 0.011).

Regarding the use of ADM, a statistically significant association was found in the prepectoral plane between their use and higher rates of global complications (*p* = 0.021) and SSI (*p* = 0.017). In retropectoral plane, ADMs were associated with higher rates of overall complications (*p* = 0.028) and surgical wound dehiscence (*p* = 0.023).

Other variables such as BMI, reconstructive plane, or expansion volume were not associated with a higher complication rate. To facilitate visual information, only the results of the univariate analysis derived from the study of the association between complications and the use of ADM, stratified by reconstructive plane, were included in the tables.

More detailed data on the association between the use of acellular matrices and the development of complications, stratified by the reconstructive plane, are presented in Table 2.

Those variables that showed statistical significance in the univariate analysis were subjected to multivariate analysis. Statistically significant associations were only found for overall complications, SSI, and surgical wound dehiscence, precluding multivariate analysis for other complications.

After multivariate analysis (Table 3 and Table 4), a statistically significant association was found between the use of acellular matrices and the development of SSI in the prepectoral plane. In the retropectoral plane, the use of acellular matrices was associated with a higher rate of overall complications and surgical wound dehiscence

Specifically, the use of acellular matrices was associated with an odds ratio (OR) of 7.62 (95% CI: 1.23–47) for SSI in the prepectoral plane. In the retropectoral plane, the use of acellular matrices was linked to an OR of 3.34 (95% CI: 1.01–11.16) for overall complications and an OR of 6.04 (95% CI: 1.02–35.76) for surgical wound dehiscence.

## 3. Discussion

This study aims to analyze outcomes and complications in two-stage expander reconstruction by plane and by the specific ADM products used in our practice, acknowledging potential confounding by indication due to surgeon preference and inherent differences between ADM products.

Although not the primary objective, consistent with previous studies, smoking and radiotherapy were identified as significant risk factors for complications in breast reconstruction [12,13,14,15,16].

The use of ADM remains controversial due to conflicting reports regarding postoperative complications. The heterogeneity of reconstructive surgical procedures, the retrospective nature of many studies, and the variety of ADM types contribute to the challenges in interpreting the benefits and risks of ADM use.

This study is among the few to compare different ADM types across two reconstructive planes. Although both groups were comparable in most clinical characteristics, NACT was significantly imbalanced across planes (30.3% vs. 9.3%; *p* = 0.012), representing a potential confounder in plane-based comparisons.

There is a paucity of studies specifically addressing ADM use in tissue expander-based breast reconstruction within the prepectoral plane. In this series, an association was found between the use of ADM and higher rates of SSI and overall complications. Poveromo et al. [15], in an uncontrolled series of 108 reconstructions with tissue expanders and ADM, reported a 19% rate of flap necrosis, 14% of SSI, and an 8% rate of explantation. Similarly, Salibian et al. [16]., in a two-stage reconstruction study (expander followed by prosthesis) involving 51 patients and 76 reconstructions, found no statistically significant difference in complications between patients who received ADM and those who did not. In our series, we observed a higher proportion of surgical wound dehiscence cases in the prepectoral plane among patients reconstructed with ADM, although the association was not statistically significant, likely due to the study’s limited sample size.

Contradictory to our findings, Pires et al. [17] found no statistically significant differences in complication rates with ADM use in a study of 124 reconstructions. Another recent non-controlled series of 46 patients reported an overall complication rate of 25.9%, with 20.4% seroma and 3.7% SSI rates, suggesting that ADM use may be a safe option [18].

The rates observed in our series are notably higher, with a 25% rate of seroma and a SSI rate of up to 50%. It is worth noting that the aforementioned study used indocyanine green to assess flap perfusion, excluding cases with poor intraoperative flap perfusion. Furthermore, the referenced study used a membrane thickness of 0.6 mm, while our series used a thickness of 1 mm. Some studies have suggested that complication rates do not differ based on the type of matrix used [19].

Researchers have explored the relationship between the thickness and origin of different ADMs and complication rates. A recent study reported that thicker ADMs were associated with a higher risk of infection (*p* = 0.0178), skin necrosis (*p* = 0.0046), and reoperation (*p* = 0.0022). Thicker ADMs were also associated with higher drainage volume after two weeks, showing a positive correlation between thickness and drainage production time [20].

Regarding the retropectoral plane, a statistically significant association was found between the use of ADM and a higher rate of surgical wound dehiscence. The literature contains numerous studies on this topic, although the findings remain contradictory.

Brunbjerg et al. [21]., in a series of 43 two-stage reconstructions (retropectoral expander followed by implant), found no difference in complication rates compared to single-stage reconstructions (retropectoral implant with ADM assistance), although the groups were not directly comparable to the present series.

Young et al. [22], in a study of 57 two-stage reconstructions with postoperative flap necrosis, reported a 28% explantation rate in the ADM-assisted reconstruction group, compared to 6.3% in the total submuscular coverage group (*p* = 0.034).

Zhao et al. [23], in a meta-analysis of 1647 reconstructions with expanders and implants, found a higher incidence of overall complications, SSI and seroma in the ADM group, although explantation rates were not statistically different.

Similarly, Lee et al. [24], in a meta-analysis of 6.199 two-stage reconstructions, reported a statistically significant association between acellular matrix use and the development of SSI, seroma, and flap necrosis, with no difference in explantation rates.

The literature remains divided on this issue, and further research is needed to confirm these associations.

Identifying potential risk factors and individual characteristics associated with higher complication rates is essential for selecting candidates for this type of reconstruction.

Some authors have suggested that fenestrated ADMs may reduce seroma risk by allowing periprosthetic exudate drainage and facilitating reabsorption, as well as tissue neovascularization. Appropriate placement of drains and prolonged maintenance could prevent the development of seroma, although this may increase the risk of infection [20,25,26].

Exposure of a non-neovascularized acellular matrix may have similar clinical implications as implant exposure, including the risk of bacterial contamination and subsequent infection, often requiring removal. Clinically, this situation often results from mastectomy flap necrosis or surgical wound dehiscence. Selecting patients with flaps exhibiting good signs of vitality, assessed clinically or with indocyanine green, is crucial for managing complications. Conversely, low-thickness acellular matrices may accelerate tissue integration, so that if dehiscence occurs, exposure of an already integrated matrix [27,28]. However, some of these studies are not directly comparable to our two-stage approach and/or incorporated ICG-based flap selection, which may lower complication rates.

Among the complications associated with ADM use, surgical site infection and scar dehiscence were most frequent. One possible explanation is that incorporating a devitalized foreign body, such as these matrices, increases the risk of bacterial colonization and subsequent infection. Furthermore, seroma, infection, and dehiscence frequently occur sequentially when one of these complications arises. Another important limitation of this study is the heterogeneity of the products used. In our series, FORTIVA^®^ (porcine, 1 mm) was employed in the prepectoral plane, while TUTOMESH^®^ (bovine, 0.6 mm) was used in the retropectoral plane. Thickness, origin, and processing vary between these matrices, and prior literature [29,30,31,32] has suggested that thicker ADM are associated with higher rates of infection, necrosis, drainage, and reoperation. Therefore, the observed differences may reflect not only the reconstructive plane, but also the specific product characteristics, particularly thickness.

Some studies have found that higher complication rates depend on ADM thickness, as thicker matrices likely require more time and higher quality flaps to integrate [29,30]. Placing thicker ADMs when the flap is devitalized may contribute to these complications, highlighting the importance of careful patient selection [31,32].

Finally, although an association was found between hormone therapy and higher complication rates in our series, most complications appeared within the first month, while hormone therapy generally started 1–3 months postoperatively, after any complications had resolved. Therefore, this association likely resulted from the study’s small sample size, as it was not statistically significant after multivariate analysis. Although not addressed in this study, previous research by our team found higher complication rates in delayed reconstruction among patients who did not discontinue hormone therapy before surgery in delayed breast reconstruction [33].

It is important to note that comparisons with previous publications in this field are often challenging. Reported series display substantial variability in surgical technique, the type and thickness of ADM used, their origin and processing methods, as well as differences in tissue expanders and adjunctive strategies such as the use of indocyanine green angiography for flap assessment. These heterogeneities make direct comparisons difficult and likely explain some of the discrepancies in complication rates across studies. Consequently, our results should be interpreted within this broader context, emphasizing the need for further standardized and adequately powered investigations.

To our knowledge, this is among the first series exploring ADM-associated complications stratified by plane in two-stage expander reconstruction. However, like many studies in this field, this study is limited by its retrospective design and small sample size, reducing its statistical power. Nevertheless, our study is retrospective and underpowered, with small subgroups and wide 95% OR CIs, which limit precision and raise the possibility of type I error due to multiple subgroup analyses.

A key limitation of our study is the surgeon-dependent use of ADM, which may introduce confounding. In our setting, however, patients were allocated from a common surgical pool and all surgeons operated on similar case mixes. Two surgeons systematically used ADM in nearly all reconstructions, while the other two consistently avoided them, reflecting a difference in surgical philosophy rather than patient selection. Thus, ADM use was determined before assessing flap quality and was not reserved for complex or compromised cases, reducing the risk of indication bias linked to case difficulty. Nevertheless, inter-surgeon variability in technique and perioperative management cannot be entirely excluded. Furthermore, the relatively small sample size—particularly in the prepectoral + ADM subgroup—led to wide confidence intervals and limited statistical power. Therefore, our underpowered study suggests an association between ADM use and higher complication rates, but these results, while striking, require validation in larger, prospective cohorts before firm conclusions can be drawn.

These findings should be viewed as hypothesis-generating and require validation in larger, controlled, ideally multicenter cohorts. Future prospective, controlled, multicenter studies are needed to validate these results and develop standardized guidelines for acellular matrix use in breast reconstruction.

## 4. Conclusions

These findings highlight the importance of individualized surgical planning, particularly concerning the use of acellular dermal matrix, which were associated with increased risks of surgical site infection, dehiscence, and global complications. Our single-center study suggests that, in specific settings, ADM use may be associated with higher odds of infection and dehiscence. Given surgeon-driven selection, product differences and small subgroup sizes, these findings should not be generalized to all ADM or planes and require validation in larger, controlled cohorts. In light of these considerations, our findings should not be generalized to acellular dermal matrices as a uniform category. Rather, our study suggests that in the specific context of prepectoral reconstruction with FORTIVA^®^ (1 mm) and retropectoral reconstruction with TUTOMESH^®^ (0.6 mm), ADM use may be associated with higher complication rates. These results, while striking, require validation in larger cohorts and should be interpreted as hypothesis-generating, pending further evidence that accounts for product-specific differences such as thickness and origin.

## Figures and Tables

**Table 1 cancers-17-03185-t001:** Baseline characteristics.

	Overall Series (N = 87)	Prepectoral (N = 34)	Retropectoral (N = 52)	*p* Value
Age	50 (24–70)	52 (24–62)	49 (25–70)	0.234
HTN	20 (23)	5 (15.2)	15 (27.8)	0.174
DLP	17 (19.5)	4 (12.1)	13 (24.1)	0.172
BMI	24 (15.5–39)	24.48 (20–39)	23.50 (15.5–34.8)	0.249
Smoking	31 (35.6)	11 (33.3)	20 (37)	0.726
DM	5 (5.7)	1 (3)	4 (7.4)	0.395
Histological classification				0.487
LCIS	1 (1.1)	-	1 (1.9)	
DCIS	9 (10.3)	2 (6.1)	7 (13)	
ILC	6 (6.9)	3 (9.1)	3 (5.6)	
IDC	65 (74.7)	27 (81.8)	38 (70.4)	
High risk lesions	6 (6.9)	1 (3)	5 (9.3)	
Molecular classification				0.435
Luminal A	36 (41.4)	13 (39.4)	23 (42.6)	
Luminal B	29 (33.39	10 (30.3)	19 (35.2)	
HER+	10 (11.5)	6 (18.2)	4 (7.4)	
Basal like	6 (6.9)	3 (9.1)	3 (5.6)	
Unclassifiable	6 (6.9)	1 (3)	5 (9.3)	
LNI	26 (29.9)	12 (36.4)	14 (25.9)	0.302
NACT	15 (17.2)	10 (30.3)	5 (9.3)	0.012
RT	34 (39.1)	14 (42.4)	20 (37)	0.617
ACT	34 (39.1)	15 (45.5)	19 (35.2)	0.341
HT	70 (80.5)	26 (78.8)	4 (81.5)	0.759
ADM				0.380
None	50 (57.5)	17 (51.5)	33 (61.1)	
Fortiva	16 (18.4)	16 (48.5)	-	
Tutomesh	21 (24.1)	-	21 (38.9)	
Plane				
Prepectoral	33 (37.9)			
Retropectoral	54 (62.1)			
Volume	400 (200–600)	400 (200–500)	400 (300–600)	0.240

HTN: hypertension; DLP: dyslipidemia; BMI: body mass index; DM: diabetes mellitus; LCIS: lobular carcinoma in situ; DCIS: ductal carcinoma in situ; ILC: invasive lobular carcinoma; IDC: invasive ductal carcinoma LNI: lymph node involvement; NACT: neoadyuvane chemotherapy; RT: adjuvant radiotherapy; ACT: adjuvant chemotherapy; HT hormonal therapy; ADM: acellular dermal matrix.

**Table 2 cancers-17-03185-t002:** Results from univariate analysis.

	Prepectoral (n = 33)			Retropectoral (n = 54)		
	ADM (16)	No ADM (17)	*p* Value	ADM (21)	No ADM (33)	*p* Value
**OC**	9 (56.3)	3 (17.6)	0.021	12 (57.1)	9 (27.3)	0.028
**Seroma**	4 (25)	1 (5.9)	0.126	3 (14.3)	4 (12.1)	0.817
**SSI**	8 (50)	2 (11.8)	0.017	5 (23.8)	3 (9.1)	0.138
**Skin/NAC necrosis**	1 (6.3)	0	0.295	4 (19)	3 (9.1)	0.288
**Dehiscence**	7 (43.8)	3 (17.6)	0.103	6 (28.6)	2 (6.1)	0.023

ADM: acellular dermal matrix; OC: overall complications; SSI: surgical site infection; NAC: nipple-areolar complex.

**Table 3 cancers-17-03185-t003:** Results from multivariate analysis in prepectoral plane.

Variable	Univariate	Multivariate	OR IC95 (Multivariate)
**OC**			
Smoking	*p =* 0.05	*p =* 0.06	-
HT	*p =* 0.03	*p =* 0.99	-
ADM	*p =* 0.021	*p =* 0.254	-
**SSI**			
Smoking	*p =* 0.05	*p =* 0.223	-
ADM	*p =* 0.017	*p =* 0.029	7.62 (1.23–47)

OC: overall complications; HT: hormonal therapy; ADM: acellular dermal matrix; SSI: surgical site infection; OR: Odds Ratio; I 95: 95% Confidence Interval.

**Table 4 cancers-17-03185-t004:** Results from multivariate analysis in retropectoral plane.

Variable	Univariate	Multivariate	OR IC95 (Multivariate)
**OC**			
LNI	*p =* 0.01	*p =* 0.99	-
ADM	*p =* 0.028	*p =* 0.05	3.34 (1.01–11.16)
**SSI**			
Smoking	*p =* 0.04	*p =* 0.05	2.45 (1.01–6.10)
RT	*p* = 0.05	*p =* 0.05	2.20 (0.98–15.02)
**Dehiscence**			
LNI	*p =* 0.011	*p =* 0.912	-
ADM	*p =* 0.023	*p =* 0.05	6.04 (1.02–35.76)

OC: overall complications; LNI: lymph node involvement; ADM: acellular dermal matrix; SSI: surgical site infection; RT: radiotherapy; OR: Odds Ratio; I 95: 95% Confidence Interval.

## Data Availability

The data that support the findings of this study are available upon reasonable request from the corresponding author. Restrictions may apply due to privacy, ethical concerns or ongoing analysis.

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
