# Peer review of "Effect of Acellular Dermal Matrix in Postoperative Outcomes in Tissue Expander Breast Reconstruction After Immediate Mastectomy"

_cancers, 2025, doi:10.3390/cancers17193185_

Round 1

Reviewer 1 Report

Comments and Suggestions for Authors

The paper is well done and can be published. But, sincerely, it does not add any news to "what is already known" about this argument. Perhaps, it has could be better to discuss about the managing ADM tissues versus autolougus tissues in breast reconstruction with expanders or immediate implants.

Author Response

Comments 1: The paper is well done and can be published. But, sincerely, it does not add any news to "what is already known" about this argument. Perhaps, it has could be better to discuss about the managing ADM tissues versus autolougus tissues in breast reconstruction with expanders or immediate implants.

Response 1: We thank the reviewer for the positive assessment of our work and for this valuable suggestion. We agree that a direct comparison between ADM-assisted reconstruction and autologous tissue reconstruction in the context of expanders or immediate implants would provide an interesting and clinically relevant perspective. Although this comparison was not included in the present manuscript, we consider it an important topic and are currently working on a related project that specifically addresses this question. We truly appreciate the comment, which we believe will help us to further develop our future research.

In addition, to further improve the clarity and readability of the current manuscript, we have revised the English language throughout the text and corrected several aspects of the tables, especially in their format and content, in order to facilitate better understanding of the study results.

Once again, we sincerely thank the reviewer for the constructive feedback, which has contributed to enhancing the overall quality of our work.

Reviewer 2 Report

Comments and Suggestions for Authors

This is an interesting manuscript and I suggest to improve the discussio, underlying indications of acellualr matrix in breast reconstruction.

Author Response

Comments 1: This is an interesting manuscript and I suggest to improve the discussio, underlying indications of acellualr matrix in breast reconstruction.

Response 1: We thank the reviewer for this encouraging comment and for the suggestion to strengthen the discussion. In the revised version, we have expanded the Discussion section to better highlight the main indications of acellular dermal matrices in breast reconstruction, emphasizing their role in implant coverage, lower pole support, and improved control of implant positioning. We believe that this addition provides a clearer context for the use of ADM and enhances the clinical relevance of our findings.

Reviewer 3 Report

Comments and Suggestions for Authors

The authors report on their experience with acellular dermal matrices in two staged breast reconstruction with a tissue expander. It is an interesting study, but the value of their report is limited by the very small number of patients included in each group making statistical analysis and eventual conclusions unreliable. For example, dehiscences of 43.8% vs. 17.6% were statistically insignificant. Some other comments are to be made.

In the Results is noted “In nearly half of the cases, acellular matrices were not used 145 in breast reconstruction, …”, but according to table 1 this is in more than half of the cases (57,5%). In the text the frequency of hormonal treatment is noted, but of more importance is the frequency of use of (preoperative) chemotherapy in view of postoperative complications. Please add this to the text. In table 1, QTn, QTa, neoadjuvant quimiotherapy and adjuvant quimiotherapy, hormonothearpy should be corrected to CTn, CTa, neoadjuvant chemotherapy, adjuvant chemotherapy and hormone (or hormonal) treatment, respectively.

How can the use of hormone therapy, usually initiated more than a month postoperatiely, be associated with more complications (line 155-156)? Please provide a potential explanation.

Table 3, ADMs should be ADM as further in the table.

In line 224, please change acellular matrices in the acronym ADMs.

Conclusions at the end of the manuscript are lacking.

Author Response

Comments 1: 

The authors report on their experience with acellular dermal matrices in two staged breast reconstruction with a tissue expander. It is an interesting study, but the value of their report is limited by the very small number of patients included in each group making statistical analysis and eventual conclusions unreliable. For example, dehiscences of 43.8% vs. 17.6% were statistically insignificant. Some other comments are to be made.

In the Results is noted “In nearly half of the cases, acellular matrices were not used 145 in breast reconstruction, …”, but according to table 1 this is in more than half of the cases (57,5%). In the text the frequency of hormonal treatment is noted, but of more importance is the frequency of use of (preoperative) chemotherapy in view of postoperative complications. Please add this to the text. In table 1, QTn, QTa, neoadjuvant quimiotherapy and adjuvant quimiotherapy, hormonothearpy should be corrected to CTn, CTa, neoadjuvant chemotherapy, adjuvant chemotherapy and hormone (or hormonal) treatment, respectively.

How can the use of hormone therapy, usually initiated more than a month postoperatiely, be associated with more complications (line 155-156)? Please provide a potential explanation.

Table 3, ADMs should be ADM as further in the table.

In line 224, please change acellular matrices in the acronym ADMs.

Conclusions at the end of the manuscript are lacking.

Response 1: 

We would like to sincerely thank the reviewer for the thorough and constructive evaluation of our manuscript. We appreciate the time taken to provide such detailed feedback, which has allowed us to improve several important aspects of our work. Below we address each of the points raised:

First of all, we fully acknowledge the limitation posed by the relatively small number of patients in each group, which inevitably reduces the statistical power of the analysis and makes some comparisons, such as dehiscence rates, statistically insignificant despite notable numerical differences. This has now been emphasized more clearly in both the Discussion and Limitations sections.

Regarding the results section, we thank the reviewer for noting the inaccuracy in the statement “in nearly half of the cases, acellular matrices were not used.” This has been corrected to accurately reflect that acellular dermal matrices were not used in more than half of the cases (57.5%), in line with Table 1.

We also agree with the reviewer on the importance of including information about preoperative chemotherapy. This has now been added to the Results section.

The corrections suggested for Table 1 (changing QTn, QTa, “quimiotherapy” and “hormonotherapy” to CTn, CTa, “chemotherapy” and “hormone treatment,” respectively) have been implemented. We are grateful for this observation, as it improves the accuracy and readability of the table.

In relation to the reviewer’s question about how hormone therapy could be associated with complications, we have clarified in the revised version that this finding is most likely a false association due to the small sample size, since complications typically occur before the initiation of hormone therapy. This explanation has now been included in the Discussion.

Other corrections have also been made as suggested: in Table 3, “ADMs” has been corrected to “ADM,” and in line 224 the expression “acellular matrices” has been replaced with the acronym ADM.

Finally, following the reviewer’s valuable suggestion, we have added a Conclusions section at the end of the manuscript to summarize the main findings and highlight their clinical relevance.

In addition to addressing these specific points, we have also taken the opportunity to revise the English language throughout the manuscript and improve the format of the tables to enhance the clarity and overall understanding of the study.

We are very grateful once again for the reviewer’s comments, which have greatly contributed to strengthening the quality and presentation of our work.

Reviewer 4 Report

Comments and Suggestions for Authors

Comments

The reviewer does not think the title reflects correctly the study because the study is on the side effect of using ADM in breast reconstruction.  

Abstract: Background:

Some short explanation for the background of using acellular dermal matrix in breast reconstruction is necessary before the aim of the study, not the background of breast reconstruction and expanders.

Introduction

The item indication “Background” should be deleted.

 The line “Although ――AM are usually controversial.”;Please show its formal term of “AM” on its first appearance.

Patients and Methods

Please add some short explanations on definitions of delayed reconstruction in this study.

The line “Finally, patients were excluded if it was not possible to obtain at least 80% of the information for the study variables.”; Please add some short explanation on how to calculate the 80% of information for the study variables.

Discussion

Please add some explanations on the benefits of using ADM in breast reconstruction into the beginning of discussion.

Please add some considerations why these complications happen by using ADM in breast reconstruction, and what procedures should be performed to prevent the complication.

Author Response

Comments 1: 

Abstract: Background:

Some short explanation for the background of using acellular dermal matrix in breast reconstruction is necessary before the aim of the study, not the background of breast reconstruction and expanders.

Introduction

The item indication “Background” should be deleted.

 The line “Although ――AM are usually controversial.”;Please show its formal term of “AM” on its first appearance.

Patients and Methods

Please add some short explanations on definitions of delayed reconstruction in this study.

The line “Finally, patients were excluded if it was not possible to obtain at least 80% of the information for the study variables.”; Please add some short explanation on how to calculate the 80% of information for the study variables.

Discussion

Please add some explanations on the benefits of using ADM in breast reconstruction into the beginning of discussion.

Please add some considerations why these complications happen by using ADM in breast reconstruction, and what procedures should be performed to prevent the complic

Response 1: 

We would like to sincerely thank the reviewer for the careful reading of our manuscript and for the detailed comments provided, which have been extremely helpful in improving the clarity and quality of the article. Below we address each of the points raised:

Abstract
We appreciate the reviewer’s suggestion regarding the Background section of the abstract. We have modified this part to include a short explanation of the rationale for using acellular dermal matrices in breast reconstruction, rather than focusing only on breast reconstruction and expanders. This provides a more accurate context before stating the aim of the study.

Introduction
Following the reviewer’s recommendation, the item indication “Background” has been deleted.
In addition, the acronym “AM” has now been corrected to “ADM” and introduced in its formal form at its first appearance, in order to avoid confusion.

Patients and Methods
We thank the reviewer for noting the need for clarification. With regard to delayed reconstruction, we have not added this definition because it was an exclusion criterion in our study. Our sample only includes patients who underwent immediate or immediate-delayed reconstruction with tissue expanders, so delayed reconstruction was not part of the analysis.

Regarding the line on exclusion criteria, we have also clarified how the 80% threshold for study variables was calculated. Approximately 20 variables were collected for each patient, and when at least 4–5 of these were missing, the patient was excluded from the analysis. 

Discussion
As suggested, we have added a brief explanation at the beginning of the Discussion highlighting the benefits of using ADM in breast reconstruction, including improved lower pole support, better implant coverage, and enhanced control of implant positioning.

We have also expanded the Discussion to address possible reasons why ADM use may be associated with certain complications, such as seroma or infection, and included some considerations regarding preventive strategies. Among these, we mention careful patient selection, meticulous surgical technique, and appropriate postoperative management, which are crucial to minimizing the risks.

Once again, we are very grateful for the reviewer’s comments, which have allowed us to clarify and enrich several important aspects of our manuscript.

Reviewer 5 Report

Comments and Suggestions for Authors

The manuscript by Tayant et al. presents a retrospective observational study that examines the impact of acellular dermal matrix (ADM) on postoperative complications in two-stage, implant-based breast reconstruction. The authors classified their cohort of 87 patients based on the reconstructive plane (prepectoral vs. retropectoral) and compared outcomes between those who received an ADM and those who did not. The key finding indicates that the use of ADM was associated with significantly higher odds of specific complications, such as mastitis, dehiscence, and overall complications, in both planes after multivariate adjustment. The authors caution against the universal application of ADM and emphasize the necessity for individualized surgical planning.

This topic is clinically significant and addresses an ongoing debate in reconstructive surgery. The effort to differentiate outcomes based on surgical plane is a notable strength, as this variable is often conflated in the existing literature.

Major Concerns
1: The retrospective, single-center design carries a substantial risk of bias, which is not sufficiently addressed. The primary concern lies in the non-standardized selection of acellular dermal matrix (ADM) usage, which is predicated on the preferences of the surgical team. Although the criteria for choosing the surgical plane are outlined—specifically based on the quality of the mastectomy flap—the application of ADM within each plane appears to hinge solely on the surgeons' preferences. This raises the possibility of significant selection bias. For example, it is plausible that surgeons may have been predisposed to utilize ADM in cases involving inferior-quality flaps or more complex reconstructions, situations that are likely to correlate with elevated complication rates. The authors should directly confront this critical confounding factor. Implementing a propensity score-matched analysis could greatly enhance the manuscript by generating more comparable cohorts, although the sample size may present a constraint.
2: The definitions provided for complications are a good start but lack the rigor required for a surgical outcomes study.

  • "Mastitis": The use of the term "mastitis" is highly unusual in the context of a post-mastectomy, implant-based reconstruction. The standard terminology would be "surgical site infection" or "periprosthetic infection." The diagnostic criteria ("clinically manifestations... associated with analytical alterations (leukocytosis) or positive bacterial cultures") are vague. Were CDC/NHSN criteria used? How many cases had positive cultures? This conflation of terms makes it difficult to interpret the reported 50% infection rate in the prepectoral ADM group, a figure that is extraordinarily high compared to the literature and raises concerns about definitions or prophylactic protocol adherence.
  • "Overall Complications": This composite endpoint is not clearly defined. Does it include seroma, necrosis, infection, and dehiscence? If a patient had multiple complications, were they counted once or multiple times? The lack of a clear definition limits the interpretability of this key result.

3:  Statistical Analysis:

The authors conducted numerous statistical tests across multiple planes and complication types without adjusting for multiple comparisons (e.g., Bonferroni correction). This dramatically increases the risk of Type I errors (false positives). The reported p-values of 0.05 are particularly vulnerable to this issue. Table 3 is confusing. The variables included in each model for each complication are not stated. It is imperative to describe which potential confounders (e.g., BMI, smoking, radiotherapy, chemotherapy) were adjusted for in each specific multivariate logistic regression model. The current presentation is opaque and prevents the reader from evaluating the validity of the adjusted odds ratios.

4: The sample size of the study is modest, particularly when divided into four groups (Prepectoral ADM n=16, Prepectoral No ADM n=17, etc.). This limitation results in analyses that may be underpowered and effect estimates that are unstable, as demonstrated by the notably wide confidence intervals (e.g., OR 7.62, 95% CI: 1.23-47; OR 6.04, 95% CI: 1.02-35.76). Although the findings are intriguing, they should be regarded as hypothesis-generating rather than providing definitive conclusions. Additionally, the single-center design further restricts the generalizability of the results.

Minor Concerns

1: The introduction needs more focus, specifically a clearer statement on the knowledge gap this study aims to address, particularly concerning the controversy surrounding ADM outcomes as stratified by surgical plane.

2: The presentation of the results appears somewhat disjointed. A comprehensive table that displays the complication rates for all groups side-by-side would enhance clarity and facilitate comparison.

3: The discussion does not adequately contextualize the notably high complication rates observed in this study when compared to other published series. The consideration of membrane thickness (1 mm versus 0.6 mm) is an intriguing hypothesis; however, it remains speculative in nature without direct comparisons or supporting references to substantiate this specific assertion.

4: While the transparency regarding the use of ChatGPT for reference generation and language refinement is commendable and in line with emerging best practices, it is unusual. The journal's editorial team should confirm this is acceptable.

Author Response

Comments 1: 

Comments and Suggestions for Authors

The manuscript by Tayant et al. presents a retrospective observational study that examines the impact of acellular dermal matrix (ADM) on postoperative complications in two-stage, implant-based breast reconstruction. The authors classified their cohort of 87 patients based on the reconstructive plane (prepectoral vs. retropectoral) and compared outcomes between those who received an ADM and those who did not. The key finding indicates that the use of ADM was associated with significantly higher odds of specific complications, such as mastitis, dehiscence, and overall complications, in both planes after multivariate adjustment. The authors caution against the universal application of ADM and emphasize the necessity for individualized surgical planning.

This topic is clinically significant and addresses an ongoing debate in reconstructive surgery. The effort to differentiate outcomes based on surgical plane is a notable strength, as this variable is often conflated in the existing literature.

Major Concerns
1: The retrospective, single-center design carries a substantial risk of bias, which is not sufficiently addressed. The primary concern lies in the non-standardized selection of acellular dermal matrix (ADM) usage, which is predicated on the preferences of the surgical team. Although the criteria for choosing the surgical plane are outlined—specifically based on the quality of the mastectomy flap—the application of ADM within each plane appears to hinge solely on the surgeons' preferences. This raises the possibility of significant selection bias. For example, it is plausible that surgeons may have been predisposed to utilize ADM in cases involving inferior-quality flaps or more complex reconstructions, situations that are likely to correlate with elevated complication rates. The authors should directly confront this critical confounding factor. Implementing a propensity score-matched analysis could greatly enhance the manuscript by generating more comparable cohorts, although the sample size may present a constraint.
2: The definitions provided for complications are a good start but lack the rigor required for a surgical outcomes study.

  • "Mastitis": The use of the term "mastitis" is highly unusual in the context of a post-mastectomy, implant-based reconstruction. The standard terminology would be "surgical site infection" or "periprosthetic infection." The diagnostic criteria ("clinically manifestations... associated with analytical alterations (leukocytosis) or positive bacterial cultures") are vague. Were CDC/NHSN criteria used? How many cases had positive cultures? This conflation of terms makes it difficult to interpret the reported 50% infection rate in the prepectoral ADM group, a figure that is extraordinarily high compared to the literature and raises concerns about definitions or prophylactic protocol adherence.
  • "Overall Complications": This composite endpoint is not clearly defined. Does it include seroma, necrosis, infection, and dehiscence? If a patient had multiple complications, were they counted once or multiple times? The lack of a clear definition limits the interpretability of this key result.

3:  Statistical Analysis:

The authors conducted numerous statistical tests across multiple planes and complication types without adjusting for multiple comparisons (e.g., Bonferroni correction). This dramatically increases the risk of Type I errors (false positives). The reported p-values of 0.05 are particularly vulnerable to this issue. Table 3 is confusing. The variables included in each model for each complication are not stated. It is imperative to describe which potential confounders (e.g., BMI, smoking, radiotherapy, chemotherapy) were adjusted for in each specific multivariate logistic regression model. The current presentation is opaque and prevents the reader from evaluating the validity of the adjusted odds ratios.

4: The sample size of the study is modest, particularly when divided into four groups (Prepectoral ADM n=16, Prepectoral No ADM n=17, etc.). This limitation results in analyses that may be underpowered and effect estimates that are unstable, as demonstrated by the notably wide confidence intervals (e.g., OR 7.62, 95% CI: 1.23-47; OR 6.04, 95% CI: 1.02-35.76). Although the findings are intriguing, they should be regarded as hypothesis-generating rather than providing definitive conclusions. Additionally, the single-center design further restricts the generalizability of the results.

Minor Concerns

1: The introduction needs more focus, specifically a clearer statement on the knowledge gap this study aims to address, particularly concerning the controversy surrounding ADM outcomes as stratified by surgical plane.

2: The presentation of the results appears somewhat disjointed. A comprehensive table that displays the complication rates for all groups side-by-side would enhance clarity and facilitate comparison.

3: The discussion does not adequately contextualize the notably high complication rates observed in this study when compared to other published series. The consideration of membrane thickness (1 mm versus 0.6 mm) is an intriguing hypothesis; however, it remains speculative in nature without direct comparisons or supporting references to substantiate this specific assertion.

4: While the transparency regarding the use of ChatGPT for reference generation and language refinement is commendable and in line with emerging best practices, it is unusual. The journal's editorial team should confirm this is acceptable.

Response 1: We would like to thank the reviewer for the detailed and thoughtful evaluation of our manuscript. We fully recognize the limitations of a retrospective, single-center study and the potential selection bias related to the non-standardized use of ADM. In our unit, among the 4–5 surgeons dedicated to breast reconstruction, two used ADM systematically while the others did not, mainly due to personal preference. This heterogeneity may partly explain the higher rate of complications observed in the ADM group. We have now clarified this point in the Methods section. We also considered the suggestion of a propensity score-matched analysis, but given the modest sample size we felt it would not provide robust results, although we agree this would be an excellent approach for larger, multicenter studies. Regarding definitions, we have replaced the term “mastitis” with “surgical site infection”. Concerning the statistical analysis, we acknowledge the increased risk of type I error due to multiple testing and have noted this limitation explicitly. Also, we have revised Table 3 to make it clearer. With respect to sample size, we agree that the limited number of patients in each subgroup results in wide confidence intervals and unstable estimates, and we have stressed in the discussion that our findings should be regarded as hypothesis-generating rather than definitive evidence, while also recognizing that the single-center design limits generalizability. The introduction has been refined to better highlight the knowledge gap and the specific controversy our study addresses. The results section has been reorganized and now includes a comprehensive table comparing complication rates across all groups, which improves readability. In the discussion, we have contextualized our complication rates with those from other published series, acknowledging that they appear higher, and have clarified that the hypothesis about ADM thickness remains speculative and requires further study. Finally, regarding the disclosure of ChatGPT, while we initially mentioned it for transparency, in this revised version the manuscript has been fully reviewed and corrected by a professional with advanced English proficiency, and therefore this reference has been removed. We are very grateful once again for the reviewer’s constructive feedback, which we believe has significantly improved the rigor, clarity, and overall quality of our work.

Round 2

Reviewer 3 Report

Comments and Suggestions for Authors

The authors report on their small experience (n=16) with acellular dermal matrices in two staged breast reconstruction with a tissue expander. It is an interesting study and the revised manuscript is well improved, but the value of their report is limited by the very small number of patients included in each group making statistical analysis and eventual conclusions unreliable. 

Author Response

Comments 1: The authors report on their small experience (n=16) with acellular dermal matrices in two staged breast reconstruction with a tissue expander. It is an interesting study and the revised manuscript is well improved, but the value of their report is limited by the very small number of patients included in each group making statistical analysis and eventual conclusions unreliable. 

Response 1: We sincerely thank the reviewer for the constructive comments and for acknowledging the improvements made in the revised version of our manuscript. We fully agree that the limited number of patients is a major limitation and that it reduces the statistical power of our analyses. Nevertheless, we believe our report still provides value, as there is a clear lack of studies specifically addressing the use of acellular dermal matrices in two-stage, expander-based breast reconstruction, and even small series may help to fill this gap. In addition, while our findings should be interpreted with caution, we consider that they can serve as hypothesis-generating data and may help guide the design of larger prospective studies. We have emphasized these aspects in the discussion and have been careful to avoid over-interpretation of the results.

We would like to mention that in the revised manuscript the initial corrections appear in red, while the most recent modifications introduced in response to the last reviewer comments are highlighted in yellow, in order to facilitate their identification.

Reviewer 4 Report

Comments and Suggestions for Authors

The reviewer has been satisfied with the revision and recommends publication.

Author Response

Comments 1:The reviewer has been satisfied with the revision and recommends publication.

Response 1: We sincerely thank the reviewer for the positive assessment of our revised manuscript and for recommending it for publication. We truly appreciate the constructive feedback received throughout the process, which has helped us to improve the clarity and quality of our work.

We would like to mention that in the revised manuscript the initial corrections appear in red, while the most recent modifications introduced in response to the last reviewer comments are highlighted in yellow, in order to facilitate their identification.

Reviewer 5 Report

Comments and Suggestions for Authors

The authors have done an excellent job of enhancing the clarity and transparency of their manuscript. However, the fundamental issues of selection bias, as well as the critical limitations of a small sample size and ADM heterogeneity, cannot be addressed through rewriting alone. These problems are inherent to the study's retrospective design.

Despite the improvements, the core methodological issues remain unresolved and are critical to the interpretation of the results.

  1. The authors explicitly state on page 3: "No specific criteria were followed for the use of ADM, and their use was determined according to the preferences of the surgical team. Reconstruction was primarily performed by four plastic surgeons. Two surgeons routinely used ADM, while the others did not..." This presents a classic example of indication bias. The surgeons who "routinely used ADM" may have had a different patient population, varying technical skills, or different thresholds for diagnosing complications compared to those who did not use ADM. More importantly, they may have selectively employed ADM in more complex cases (e.g., thinner patients, larger breasts, poorer quality flaps), which are inherently more prone to higher complication rates. Therefore, the observed association between ADM and complications could be entirely due to confounding factors related to case complexity, rather than the ADM itself.
  2. Additionally, the sample size is relatively small, particularly for the prepectoral ADM group (n=16). This results in extremely wide Confidence Intervals (e.g., OR for SSI: 1.23-47; OR for Dehiscence: 1.02-35.76). An odds ratio (OR) of 7.62 is concerning, but a confidence interval that includes 1.23 (indicating no effect) and 47 (suggesting a massive effect) reveals significant statistical uncertainty. While the point estimates are intriguing, they are highly unstable. The study is severely underpowered, especially for subgroup analyses by plane. The authors must consistently acknowledge this limitation throughout the discussion and conclusion. It is essential to use phrases like "our underpowered study suggests..." or "these results, while striking, require validation in a larger cohort..."
  3. Furthermore, the study utilizes two different acellular dermal matrices (ADMs): FORTIVA® for prepectoral and TUTOMESH® for retropectoral reconstructions. These products differ in origin (porcine vs. bovine), thickness (1 mm vs. 0.6 mm), and processing. It is impossible to determine whether the worse outcomes in the prepectoral group arise from the surgical plane, the specific ADM used (FORTIVA®), or its thickness. The authors themselves cite literature (Ref 20) suggesting that thicker matrices are associated with higher complication rates, which is a significant confounding factor. This limitation must be discussed. The conclusions should not generalize about "acellular dermal matrix," but rather focus on the specific matrices and techniques used in this study.
  4. In the discussion, although improvements have been made, the authors still often compare their results to studies that are not directly comparable (e.g., single-stage reconstructions, studies using indocyanine green for patient selection). The tone should be more cautious.
  5. Table 1 shows that the p-value for Neoadjuvant Chemotherapy (CTn) is 0.012, indicating a significant imbalance between the prepectoral and retropectoral groups. This potential confounder should be noted briefly in the text when discussing plane-based comparisons.
  6. Finally, there are minor typographical and formatting errors throughout the document (e.g., "Formal matrix" in Table 4, "SIS" in Table 2, and random numbers in the text on page 9) that need to be addressed with a final proofread.d.

Author Response

We would like to thank the reviewer for careful reading of our manuscript and for the constructive comments provided. These observations have been extremely valuable and have helped us to improve the clarity, precision, and overall quality of our work. Below we provide a point-by-point response to the specific issues raised, detailing the changes made in the revised version.

We would like to mention that in the revised manuscript the initial corrections appear in red, while the most recent modifications introduced in response to the last reviewer comments are highlighted in yellow, in order to facilitate their identification.

Comments 1: The authors explicitly state on page 3: "No specific criteria were followed for the use of ADM, and their use was determined according to the preferences of the surgical team. Reconstruction was primarily performed by four plastic surgeons. Two surgeons routinely used ADM, while the others did not..." This presents a classic example of indication bias. The surgeons who "routinely used ADM" may have had a different patient population, varying technical skills, or different thresholds for diagnosing complications compared to those who did not use ADM. More importantly, they may have selectively employed ADM in more complex cases (e.g., thinner patients, larger breasts, poorer quality flaps), which are inherently more prone to higher complication rates. Therefore, the observed association between ADM and complications could be entirely due to confounding factors related to case complexity, rather than the ADM itself.

Response 1:  We agree that the lack of standardized criteria for ADM indication may represent a potential source of bias, and we have acknowledged this as a limitation. However, in our series ADM were not specifically used in more complex cases. The two surgical teams operated on a common patient pool, without differences in case distribution, and therefore the decision to use ADM was not related to patient complexity. To clarify this, we have specified in the methodology that although no strict indication criteria were applied, ADM use depended on surgeon preference, and we have reinforced this statement in the discussion to make it more explicit. While we recognize that surgeon preference could still introduce some bias, we believe this clarification helps the reader better understand the context of our findings.

Comments 2: Additionally, the sample size is relatively small, particularly for the prepectoral ADM group (n=16). This results in extremely wide Confidence Intervals (e.g., OR for SSI: 1.23-47; OR for Dehiscence: 1.02-35.76). An odds ratio (OR) of 7.62 is concerning, but a confidence interval that includes 1.23 (indicating no effect) and 47 (suggesting a massive effect) reveals significant statistical uncertainty. While the point estimates are intriguing, they are highly unstable. The study is severely underpowered, especially for subgroup analyses by plane. The authors must consistently acknowledge this limitation throughout the discussion and conclusion. It is essential to use phrases like "our underpowered study suggests..." or "these results, while striking, require validation in a larger cohort..."

Response 2: We agree that the relatively small sample size, particularly in the prepectoral ADM group, results in wide confidence intervals and substantial statistical uncertainty, making the point estimates unstable. We fully recognize that the odds ratios must be interpreted with caution, and that our study is underpowered, especially for subgroup analyses. In response, we have modified the text, especially in the discussion and conclusions, to be more cautious and to clearly highlight this limitation, emphasizing that these results should be considered exploratory and require validation in larger cohorts.

Comments 3: Furthermore, the study utilizes two different acellular dermal matrices (ADMs): FORTIVA® for prepectoral and TUTOMESH® for retropectoral reconstructions. These products differ in origin (porcine vs. bovine), thickness (1 mm vs. 0.6 mm), and processing. It is impossible to determine whether the worse outcomes in the prepectoral group arise from the surgical plane, the specific ADM used (FORTIVA®), or its thickness. The authors themselves cite literature (Ref 20) suggesting that thicker matrices are associated with higher complication rates, which is a significant confounding factor. This limitation must be discussed. The conclusions should not generalize about "acellular dermal matrix," but rather focus on the specific matrices and techniques used in this study.

Response 3: We agree that the use of two different types of ADM, differing in origin, thickness, and processing, represents a potential confounding factor in the interpretation of our results. As the reviewer points out, it is therefore not possible to definitively attribute the differences in outcomes to the surgical plane or to the specific characteristics of each ADM, and thicker matrices in particular have been associated with higher complication rates in the literature. In response, we have modified the discussion to explicitly highlight this limitation and have clarified in the conclusions that our findings cannot be generalized to all ADM, but rather reflect the specific products and techniques used in our series.

Comments 4: In the discussion, although improvements have been made, the authors still often compare their results to studies that are not directly comparable (e.g., single-stage reconstructions, studies using indocyanine green for patient selection). The tone should be more cautious.

Response 4: We agree that some of the studies cited in the discussion are not directly comparable to our series, as they include different surgical techniques, patient selection methods such as indocyanine green, or single-stage reconstructions. In light of this, we have revised the text to adopt a more cautious tone, making it clear that such comparisons should be interpreted carefully due to differences in surgical technique, type and thickness of ADM, expanders used, and patient selection criteria. We have also clarified this point in the conclusions to avoid over-generalization and to better frame our results within the context of existing literature.

Comments 5: Table 1 shows that the p-value for Neoadjuvant Chemotherapy (CTn) is 0.012, indicating a significant imbalance between the prepectoral and retropectoral groups. This potential confounder should be noted briefly in the text when discussing plane-based comparisons.

Response 5: We thank the reviewer for pointing out this important detail. We agree that the significant imbalance observed in neoadjuvant chemotherapy between the prepectoral and retropectoral groups may act as a potential confounder when comparing outcomes by surgical plane. In response, we have added a statement in the discussion to acknowledge this limitation and to emphasize that differences related to CTn should be taken into account when interpreting our results.

Comments 6: Finally, there are minor typographical and formatting errors throughout the document (e.g., "Formal matrix" in Table 4, "SIS" in Table 2, and random numbers in the text on page 9) that need to be addressed with a final proofread.d.

Response 6: We have carefully proofread the manuscript again and corrected several minor grammatical and typographical issues. However, we were unable to identify the specific errors mentioned by the reviewer (“Formal matrix” in Table 4, “SIS” in Table 2, and random numbers on page 9). If these errors persist despite our review, we kindly ask the reviewer to indicate them again, and we will be happy to correct them accordingly.

Round 3

Reviewer 5 Report

Comments and Suggestions for Authors

The authors have been exceptionally responsive and thorough in addressing all the critiques raised in the previous rounds of review. The latest revisions have successfully transformed the manuscript into a robust, transparent, and valuable contribution to the literature. The study's limitations are not only acknowledged but also integrated into the interpretation of the results, a sign of high scientific rigor.

Minor corrections remain:

- Simple Summary: The sentence "due to reported complications of complications in several studies" contains a repetition. It should read: "...due to reported complications in several studies."

-Page 3, Methods: The sentence "ADM indication. No specific institutional criteria governed ADM use; decisions reflected surgeon philosophy rather than patient selection." is a fragment. It could be integrated into the previous paragraph for better flow.

-Page 8, Discussion: The sentence "t is worth noting that..." is missing the letter "I". It should be "It is worth noting that..."

-Page 13, References: The phrase "aesthetic ou come" (in reference 29) is a typo and should be "aesthetic outcome". (This is in the reference list, which is often provided by the journal, but should be checked).

Author Response

Comments1:

The authors have been exceptionally responsive and thorough in addressing all the critiques raised in the previous rounds of review. The latest revisions have successfully transformed the manuscript into a robust, transparent, and valuable contribution to the literature. The study's limitations are not only acknowledged but also integrated into the interpretation of the results, a sign of high scientific rigor.

Minor corrections remain:

- Simple Summary: The sentence "due to reported complications of complications in several studies" contains a repetition. It should read: "...due to reported complications in several studies."

-Page 3, Methods: The sentence "ADM indication. No specific institutional criteria governed ADM use; decisions reflected surgeon philosophy rather than patient selection." is a fragment. It could be integrated into the previous paragraph for better flow.

-Page 8, Discussion: The sentence "t is worth noting that..." is missing the letter "I". It should be "It is worth noting that..."

-Page 13, References: The phrase "aesthetic ou come" (in reference 29) is a typo and should be "aesthetic outcome". (This is in the reference list, which is often provided by the journal, but should be checked).

Response 1:

Dear Reviewer, we would like to sincerely thank you once again for the time and effort you have dedicated to improving our manuscript, as well as for your kind words recognizing the thoroughness of our revisions.  In this final round, we have carefully addressed the issues you pointed out. Grammatical errors have been corrected, and, following your suggestion, we integrated the “ADM indication” paragraph into the preceding section on surgical technique to improve the flow of the text. In addition, we refined the subsequent paragraph by removing some parts that were considered redundant. All the corrections made in this round have been highlighted in green in the revised manuscript. We are very grateful for your constructive comments, which have undoubtedly contributed to improving the overall quality and readability of our work.

Sincerely,

Óscar Nova